# Proving the Superiority of Intraoperative Recurrent Laryngeal Nerve Monitoring over Visualization Alone during Thyroidectomy

**DOI:** 10.3390/biomedicines11030880

**Published:** 2023-03-13

**Authors:** Beata Wojtczak, Dominik Marciniak, Krzysztof Kaliszewski, Krzysztof Sutkowski, Mateusz Głód, Jerzy Rudnicki, Marek Bolanowski, Marcin Barczyński

**Affiliations:** 1Department of General, Minimally Invasive and Endocrine Surgery, Wroclaw Medical University, Borowska Street 213, 50-556 Wroclaw, Poland; 2Department of Dosage Form Technology, Wroclaw Medical University, Borowska Street 211 A, 50-556 Wroclaw, Poland; 3Department of Endocrinology, Diabetes and Isotope Therapy, Wroclaw Medical University, Pasteura Street 4, 50-367 Wroclaw, Poland; 4Department of Endocrine Surgery, Third Chair of General Surgery, Jagiellonian University Medical College, 50 Mikolaja Kopernika Street, 31-501 Krakow, Poland

**Keywords:** thyroidectomy, vocal fold paresis, intraoperative nerve monitoring, risk factors, recurrent laryngeal nerve

## Abstract

Vocal fold paralysis after thyroid surgery is still a dangerous complication that significantly reduces patients’ quality of life. Since the intraoperative neuromonitoring (IONM) technique has been introduced and standardized, the most frequently asked question is whether its use has significantly reduced the rate of RLN injury during thyroid surgery compared to visual identification alone (VA). The aim of this study was to attempt to prove the superiority of IONM over VA of the RLN during thyroid surgery in the prevention of vocal fold paralysis, taking into account risk factors for complications. The medical records of 711 patients (1265 recurrent laryngeal nerves at risk of injury) were analyzed retrospectively: in 257 patients/469 RLNs at risk, thyroid surgery was performed with IONM; in 454 patients/796 RLNs at risk, surgery was performed with VA. The statistical analysis showed that in the group of patients with IONM only one risk factor—the surgeon’s experience—proved statistically significant (OR = 3.27; *p* = 0.0478) regarding the overall risk of vocal fold palsy. In the group of patients where only visualization was used, 5 of the 12 factors analyzed were statistically significant: retrosternal goiter (OR = 2.23; *p* = 0.041); total thyroid volume (OR = 2.30; *p* = 0.0284); clinical diagnosis (OR = 2.5; *p* = 0.0669); gender (OR = 3.08; *p* = 0.0054) and risk stratification (OR = 3.30; *p* = 0.0041). In addition, the cumulative risk, taking into account the simultaneous influence of all 12 factors, was slightly higher in the group of patients in whom only VA was used during the procedure: OR = 1.78. This value was also considerably more statistically significant (*p* < 0.0001) than that obtained in the group of patients in whom IONM was used: OR = 1.73; *p* = 0.004. Conclusions: Risk factors for complications in thyroid surgery are not significant for any increase in the rate of vocal fold paralysis as long as surgery is performed with IONM, in contrast to thyroid surgery performed only with VA, thus proving the superiority of IONM over VA for safety.

## 1. Introduction

“The extirpation of the thyroid gland for goiter typifies perhaps better than any other operation the supreme triumph of the surgeon’s art”. This 1920 quote from William Halstedt clearly encapsulates the progress made in thyroid surgery in the early 20th century [1]. Thyroid surgery, which had previously carried a 40% mortality risk and been banned by the French Academy of Medicine, had finally become safe, with a mortality rate of less than 0.5% [2]. However, the rate of vocal fold paralysis remained high and went unchanged for decades to come, until 1938, when Lahey pointed out that identifying the recurrent laryngeal nerve (RLN) during thyroid surgery definitely reduces its injury rate, and this discovery began a new era in thyroid surgery [3]. Today, no one doubts that identifying the RLN during thyroid surgery is the “gold standard”, which was confirmed, among other things, in a multicenter study by Jatzko et al. [4] On the basis of more than 12,000 thyroid surgeries, Jatzko et al. showed that the rates of transient and permanent vocal fold paralysis among patients without RLN identification (7.9% and 5.2%, respectively) were statistically significantly higher than in the group of patients with visual RLN identification, where the complication rates were 2.7% and 1.2% [4]. The introduction of intraoperative neuromonitoring (IONM) of the RLN during thyroid surgery in 1966 was another milestone in thyroid surgery that further reduced the rate of vocal fold paralysis [5]. The IONM technique uses electromyography of the vocal folds to monitor the electrophysiological response from the RLNs when they are stimulated during thyroid surgery. However, for many years, due to the relatively poor quality of nerve monitoring equipment and its lack of standardization, the technique did not enter widespread use. It was not until 2011 that the International Neuromonitoring Group’s guidelines for RLN monitoring were first published [6], followed in 2013 by guidelines for monitoring the external branch of the superior laryngeal nerve (EBSLN) [7], which led to a rapid increase in interest in the technique of laryngeal nerve monitoring during thyroid surgery among both surgeons and otolaryngologists worldwide [8,9]. Currently, intermittent neuromonitoring (I-IONM), which involves direct stimulation of the RLN during thyroid surgery via a mono- or bipolar probe, and continuous neuromonitoring (C-IONM), which involves automatic continuous stimulation via a probe placed directly on the vagus nerve, are both used [10]. The main advantage of neuromonitoring over visualization alone (VA) is the ability to assess not only the anatomical continuity of the RLN, but also its functional integrity during thyroid surgery itself. This provides the opportunity to opt for an appropriate strategy during thyroid surgery: a so-called “staged thyroidectomy” to avoid bilateral vocal fold paresis [6,11]. Further advantages of IONM include the ability to identify more than 98% of RLNs even in a scarred surgical field, the possibility of intraoperative recognition of the type of RLN injury and facilitation of isolating the RLN from the surrounding tissues. In addition, IONM enables recognition of anatomical variants of the RLN, plays an educational role for the surgical team and may be used for legal reasons [6,7,8,9,10,11]. However, the main purpose of introducing IONM into thyroid surgery was to prevent RLN injury during thyroid surgery. Thus, for more than two decades, the question has been whether IONM has reduced the frequency of vocal fold paralysis to a statistically significant degree compared with surgery with only visual identification of the RLN. The results of papers from single centers, multicenter papers and meta-analyses are quite divergent on this topic. In most published works, the percentage of vocal fold paralysis in thyroid surgery with IONM is lower than with VA, but the differences do not reach the level of statistical significance [12,13,14,15], which seriously limits the possibility of demonstrating the superiority of IONM over VA. To date, only a few meta-analyses, including those by Zheng et al. [16], Yang et al. [17], Wong et al. [18] and Bai et al. [19], have shown an advantage of IONM over VA, but only in specific groups of patients and specific types of vocal fold paralysis. The question therefore remains: can we prove the advantage of IONM over VA in the prevention of RLN injury during thyroid surgery?

The aim of this study was to attempt to prove the superiority of IONM over visual identification of the RLN during thyroid surgery in the prevention of vocal fold paralysis, taking into account risk factors for complications. We aimed to prove the hypothesis that commonly known risk factors for complications do not have a significant impact on the incidence of vocal fold paralysis in a group of patients operated on with IONM, in contrast to patients operated on with VA.

## 2. Materials and Methods

On 13 May 2020, the consent of the Bioethics Committee of the Medical University of Wroclaw to conduct the study was obtained (KB-280/2020, 13 May 2020).

### 2.1. Study Population

The medical records of 711 patients (1265 recurrent laryngeal nerves at risk of injury) who were surgically treated for various disorders of the thyroid gland at the Department of General, Gastroenterological and Endocrine Surgery of the Medical University of Wroclaw in 2012–2015 were analyzed retrospectively (as of 2018, the name of the Department has changed; the current name is the Department of General, Minimally Invasive and Endocrine Surgery of the Medical University of Wroclaw). In 257 patients (469 RLNs at risk), thyroid surgery was performed with intraoperative neuromonitoring (IONM), and in 454 patients (796 RLNs at risk), only visual identification of the RLN was used during thyroid surgery. For the first 4 months of each year, patients were operated on with IONM, and in the following months operations were performed without IONM, only with VA. The demographic data, clinical data and characteristics of the patients operated on with IONM and with VA are shown in Table 1. There were no statistically significant differences between the 2 groups in terms of age, gender, the number of patients with tracheal displacement/tracheal stenosis, retrosternal goiter, type of surgery, or operator experience (*p* > 0.05).

### 2.2. Study Design and Definitions

In order to verify the research hypothesis, we planned to carry out the following:Compare the number of occurrences of vocal fold paralysis: total, transient and permanent in the groups of patients operated with IONM vs. VA.Compare the total number of occurrences of vocal fold paralysis in each subgroup of patients, taking into account the risk factors for complications.Evaluate the influence of the risk factors for complications on the occurrence of vocal fold paralysis in the group of patients operated on with IONM and separately in the group of patients operated on with VA.Then, in order to statistically evaluate the effect of the use of neuromonitoring on the risk of postoperative complications in the 2 groups of patients, 2 multigroup variable effect meta-analysis models were constructed using the odds ratio (OR) along with a 95% significance interval as the test metric. In both cases, the total risk of postoperative complications was determined by the values of 12 dichotomous variables, which are shown in Table 2.

Risk factor definitions are as follows:(1)Age definition: age ≤ 64 years—patients up to 64 years of age. Age > 64 years—patients aged 65 and over.(2)Gender: female or male.(3)BMI: BMI < 25 (kg/m^2^) or BMI ≥ 25 (kg/m^2^).(4)Clinical diagnosis: benign goiter (nodular goiter, toxic nodular goiter and Graves’ disease). Carcinoma (papillary, follicular, medullary and anaplastic). Patients with thyroid cancer underwent thyroid operation with central lymph node dissection.(5)Thyroid function: euthyroidism (patients with nodular goiter without thyroid dysfunction before surgery). Thyrotoxicosis (patient treated before thyroid operation due to thyrotoxicosis; patients with toxic nodular goiter or Graves’ disease)(6)Focal lesion: single (single thyroid nodule) or multiple (multiple thyroid nodule) according to ultrasound examination.(7)Total volume: the volume of the thyroid lobe determined on the basis of the formula for the volume of the volume of one lobe = length × depth × width × π/6; ≤40 mL (thyroid volume up to 40 mL) or >40 mL (thyroid volume from 40 mL and above).(8)Trachea (based on X-rays of the neck and chest): normal/constricted.(9)Retrosternal goiter (based on x-rays of the neck and chest): absent/present.(10)Surgeon experience: low volume (>100 s/y)—surgeon performing up to 100 operations per year. High volume (≤100 s/y)—surgeon performing over 100 operations a year.(11)Thyroid surgery: primary (first thyroid operation on the thyroid gland) or secondary (recurrent goiter or radicalization due to cancer).(12)Risk stratification: low-risk operation (patients operated due to nodular goiter and toxic nodular goiter) or high-risk operation (patient operated due to Graves’ disease and thyroid cancer).(13)Extent of thyroid surgery: total thyroid lobectomy (the entire thyroid lobe is removed) or partial thyroid lobectomy (a part of thyroid lobe is removed).

This study’s primary endpoint was vocal cord palsy (VCP) due to injury to the recurrent laryngeal nerve during thyroid surgery. All patients who underwent thyroid operation were followed for 12 months after surgery to assess complications.

The number of occurrences of vocal fold palsy was calculated in terms of the number of RLNs at risk of injury. To assess the mobility of the vocal folds, both preoperatively and postoperatively (2 days to 2 weeks after surgery, and 1 year later) each patient underwent an ENT examination (video laryngoscopy or indirect laryngoscopy) to assess vocal fold mobility. Transient vocal fold paralysis (T-VCP) was defined as paralysis resolving up to 12 months after thyroid surgery; permanent paralysis (P-VCP) was defined as paralysis persisting 12 months and more after surgery. Total/overall paralysis (O-VCP) was defined as the sum of all paralysis: transient and permanent, assessed in the immediate post-thyroid surgery period. Patients with preoperative vocal fold paralysis were excluded from the study. The endpoint in the analyses by risk factors was the total number of vocal fold paralysis cases found in the immediate postoperative period; separate analyses were not performed for transient and permanent paralysis due to the relatively small number of these complications not meeting the minimum value for statistical analyses.

In all the operations with IONM, RLN monitoring was carried out according to the recommendations of the International Neural Monitoring Study Group [6], employing a NIM-3.0 nerve monitor (Medtronic, Jacksonville, USA) and the intermittent IONM technique. A monopolar stimulating probe was used for nerve stimulation with a current amplitude of 1 mA (range 0.5–1.5 mA) and 3 Hz pulses of 200 ms each for 1–2 s.

### 2.3. Statistical Analysis

A series of statistical analyses was carried out on data collected from the 711 patients who constituted the study sample.

The basis of all the analyses carried out was a spreadsheet experiment matrix developed from the source data, in which the columns were the successive variables analyzed and the rows were the individual values determined for successive study participants. The variables subjected to statistical analysis were on both nominal-dichotomous and quotient scales. Some of the statistical analyses conducted used quotient variables subjected to categorization into dichotomous variables.

The statistical analysis began with verification of the correctness of the data contained in the compiled database. For this purpose, descriptive statistics with histograms were determined for quotient variables and tables of counts for nominal variables. Subsequently, all extreme outliers and obviously questionable results were again verified with the source materials. The few deficiencies present in the database were eliminated and not replaced with any other values.

In order to characterize the quantitative variables, basic descriptive statistics were calculated for them: histogram, mean value, standard deviation, range and 95% confidence intervals (±95% CI) for the mean value. The normality of the distributions of the quantitative variables was assessed with a W Shapiro-Wilk test, and the homogeneity of variance was assessed with a Levene test and Brown–Forsyth test assuming a significance level of ⍺ = 0.05. For the variables on nominal-dichotomous scales, tables of counts containing absolute raw counts and the percentage contribution of each category to the nominal variable were determined.

Multivariate analyses—principal component analysis (PCA) and correspondence analysis (CA)—based on dimension reduction using procedures for decomposing the matrix of results according to singular values were used to tentatively identify the overall relationships among all the variables being statistically evaluated. The PCA models were evaluated using the nonlinear iterative partial least squares (NIPALS) iterative algorithm with the convergence criterion set at 0.00001 and the maximum number of iterations set at 100. The number of principal components was established by determining the maximum of the predictive ability Q2 using the V-fold cross-check method, setting the maximum number at Vmax = 7. The optimal PCA model obtained was visualized graphically by plotting the 2 principal components with the largest percentage contribution to the variance described by the model (PC1 vs. PC2). The PCA analysis that was carried out, the results of which are shown in the PC1 and PC2 loadings graph, made it possible to pre-select the variables with the most significant impact on the model built and to select the most significant correlations between them. The variables selected in this way were then subjected to further detailed statistical evaluation.

Pearson’s non-parametric chi^2^ test was used to assess the statistical significance of correlations among the variables on the nominal scales; additionally, odds ratios (ORs) were determined based on the 2 × 2 bivariate tables constructed, along with 95% significance intervals (±95% CI). The results of the Pearson’s chi^2^ test of the comparisons were further subjected to pooled statistical evaluation by conducting a meta-analysis. A multigroup variable effects model of the meta-analysis was applied using the OR along with a 95% significance interval as the metric tested. The meta-analysis was further supplemented by performing a sensitivity analysis, which allowed us to separately assess the effect of each analyzed variable on the statistical significance of the constructed meta-analysis models. The results of the meta-analysis were visualized in standard forest plots.

A significance level of α = 0.05 was assumed in all the statistical analyses performed. The statistical analyses were carried out using STATISTICA PL ^®^ software (version 13.3) with the Set Plus add-on (version 3.0).

## 3. Results

### 3.1. Total, Transient and Permanent Vocal Fold Paralysis in Patients Operated on with IONM vs. VA

Table 3 shows the total, transient and permanent number of occurrences of vocal fold paralysis in the group of patients operated on with IONM vs. VA. The percentage of all types of paralysis in the group of patients operated on with IONM was lower than in the group with VA; however, our analysis by Pearson’s non-parametric chi^2^ test showed no statistical significance in the incidence of this complication in the two groups of patients (*p* > 0.05).

### 3.2. Total Cases of Vocal Fold Paralysis vs. Risk Factors for Complications in Patients Undergoing Thyroid Surgery with IONM vs. VA

Table 4 shows the total number of incidents of vocal fold paralysis with risk factors for complications in patients undergoing thyroid surgery with IONM vs. VA. In the subgroups selected on the basis of risk factors, there were no statistically significant differences in the number of incidents of vocal fold paralysis in operations with IONM vs. VA (*p* > 0.05). The subgroup of patients at so-called “high surgical risk”, which included patients with thyroid cancer and Grave’s disease, was the only subgroup in which our analysis by Pearson’s non-parametric chi^2^ test was close to reaching statistical significance (*p* = 0.0666, OR 0.23). In this subgroup of “high surgical risk” patients with IONM, the complication rate was 2.22%, while among the patients undergoing surgery with VA it was 8.91%.

### 3.3. The Effects of Risk Factors on the Total Number of Cases of Vocal Fold Paralysis in the Group of Patients Operated on with IONM and in those Operated on with VA

Table 5 shows the effects of risk factors for complications on the total number of instances of vocal fold paralysis among patients operated on with IONM as opposed to those operated on with VA.

In the group of patients operated on with IONM, our analysis with Pearson’s non-parametric chi^2^ test showed that out of 13 risk factors for complications, only one—the surgeon’s experience—had a statistically significant effect on the frequency of vocal fold paralysis in the immediate postoperative period. Among the thyroid surgeries performed with IONM by the group of surgeons with less experience, the rate of paralysis was 5.75%, which was significantly higher (*p* = 0.0478) than in the group of highly experienced surgeons (1.83%). None of the other listed risk factors for complications had statistically significant effects on the rate of vocal fold paralysis (*p* > 0.05) in patients operated on with neuromonitoring.

In the group of patients operated on with VA only, the analysis by Pearson’s non-parametric chi^2^ test showed that as many as 4 of the 13 risk factors for complications had statistically significant effects on the rate of vocal fold paralysis: male sex (*p* = 0.0054, OR 3.08), so-called “high-risk” procedures (*p* = 0.0041, OR 3.3), goiter volume above 40 mL (*p* = 0.0284, OR 2.3) and retrosternal goiter (*p* = 0.0411, OR 2.23). A fifth risk factor—thyroid cancer—came close to the level of statistical significance (*p* = 0.0669, OR 2.55).

The non-parametric analysis using Pearson’s chi^2^ test showed no effects of any of the other risk factors, such as type and extent of thyroid surgery, age, BMI, thyroid function, type of focal lesions, tracheal compression or surgeon’s experience, on the incidence of vocal fold paralysis in the group of patients who underwent thyroid surgery with VA (*p* > 0.05).

### 3.4. Meta-Analysis of the Effects of Variables and the Principal Component Analysis (PCA)

Figure 1 shows our meta-analysis determining the chances of 1 of the 12 risk factors occurring in the two groups of patients: those who had neuromonitoring applied during surgery and those in whom visual identification alone was used. The statistical analysis showed that in the group of patients in whom neuromonitoring was used, the chances of only one risk factor—the surgeon’s experience—proved statistically significant (*p* = 0.0478, OR = 3.27). Summarizing the overall risk, taking into account the simultaneous influence of all 12 risk factors in the group of patients in whom IONM was used during surgery, we obtained the following results: OR = 1.75, statistical significance *p* = 0.004 and confidence interval 1.2–2.57.

Among the patients in whom only visualization was performed, the chances of 5 of the 12 factors were statistically significant: retrosternal goiter (OR = 2.23; *p* = 0.041), total volume (OR = 2.30; *p* = 0.0284), clinical diagnosis (OR = 2.5; *p* = 0.0669), gender (OR = 3.08; *p* = 0.0054) and risk stratification (OR = 3.30; *p* = 0.0041). Summarizing the overall risk, taking into account the simultaneous influence of all 12 risk factors in the group of patients in whom only visual identification was used during the procedure, we obtained the following results: OR = 1.78, statistical significance *p* < 0.0001 and confidence interval 1.39–2.28.

Figure 2 shows the model of a meta-analysis determining the chance of early postoperative complications in the group of patients who underwent neuromonitoring during surgery that additionally took into account the presence of 1 of the 12 risk factors considered. This analysis showed that in each of the 12 subgroups, the chances of early postoperative complications among patients with whom neuromonitoring was used were lower: the OR value for each of the 12 risk factors was less than 1. Although the calculated individual OR values did not achieve statistical significance, the cumulative chances of early postoperative complications taking into account the simultaneous influence of all 12 factors did reach statistical significance: OR = 0.70; *p* = 0.016.

## 4. Discussion

Vocal fold paralysis after thyroid surgery is still a dangerous complication that significantly reduces patients’ quality of life [20]. Based on 27 articles covering 25,000 patients, Jeannon et al. [21] showed that after thyroid surgery the average rate of vocal fold paralysis, both transient and permanent, was 9.8% (2.3–26%), depending on the timing and the type of ENT examination. They noted that as many as 1 in 10 patients experience transient paralysis after thyroid surgery, and that in 1 out of 25 cases, the voice problems are permanent [21]. Hence, any measures leading to minimization of the risk of vocal fold injury seem particularly important to improve the surgical treatment of thyroid gland disorders. Visual identification of the RLN is considered the safest method to prevent its injury during thyroid and parathyroid surgery [22]. The German Association of Endocrine Surgeons [23], the Australian College of Surgeons [24], the International Intraoperative Neural Monitoring Study Group [6] and the American Academy of Otolaryngology Head and Neck Surgery [20] recommend the use of IONM in addition to visual identification of the RLN to further minimize the risk of RLN injury during thyroid surgery. Since the IONM technique has been introduced and standardized, the most frequently asked question is whether its use has significantly reduced the rate of RLN injury during thyroid surgery compared to visual identification. The role and place of IONM in thyroid surgery is still debated, thus, in the literature of the last two decades, we find many publications from single centers as well as multicenter studies, meta-analyses or randomized clinical trials aimed at establishing whether there is any advantage to IONM over VA in preventing RLN injury during thyroid surgery [12,13,14,15,16,17,18,19,22,25,26,27].

The vast majority of publications indicate that IONM reduces the risk of RLN injury compared to visualization alone during thyroid surgery, but the results obtained are often not at the level of statistical significance [12,13,14,15,25,26]. A 2022 meta-analysis by Davey et al. based on eight randomized clinical trials (2521 patients/4977 RLNs at risk) showed no statistically significant difference in the reduction in total, transient and permanent RLN injury during thyroid surgery with IONM as opposed to VA (*p* = 0.069; *p* = 0.110; *p* = 0.571), although the number of these complications was always lower in the group of patients operated on with IONM [12]. These results are consistent with previous meta-analyses including only randomized trials [15,25]. The 2013 meta-analysis by Sanabria et al., based on six randomized clinical trials (1602 patients/3064 RLNs at risk), did not show a statistically significant reduction in paralysis in surgery with IONM [25], nor did the 2019 meta-analysis by Cirocchi et al., including five randomized trials (1558 patients), which also failed to confirm the superiority of IONM over VA in preventing RLN injury [15]. Subsequent large meta-analyses including not only randomized trials but also reports from single centers have not shown any advantage of IONM over VA [13,14,26]. One of the largest meta-analyses, by Pisanu et al. in 2014, based on 23,512 patients (11,475 RLNs at risk) showed less vocal fold paralysis in operations with IONM vs. VA (total paralysis rate: 3.47% vs. 3.67%; transient paralysis: 2.62% vs. 2.72%, permanent paralysis: 0.79% vs. 0.92%); however, none of these results were at the level of statistical significance [13]. A later, even larger meta-analysis by Malik et al. in 2016 [26] comparing IONM with VA was based on 17 publications and involved 30,926 patients, and a comprehensive meta-analysis by Henry et al. [14] in 2017 ultimately failed to confirm the superiority of IONM over VA in preventing RLN injury during thyroid surgery. To date, only a few publications have demonstrated the superiority of IONM over VA in minimizing the risk of vocal fold paralysis in selected groups of patients [16,17,18,19,27]. In 2009, Barczynski et al. were among the first to demonstrate a statistically significant lower rate of transient vocal fold paralysis based on the first randomized clinical trial of IONM [27]. Four years later, a meta-analysis by Zheng et al. based on 14 studies involving 36,487 patients confirmed Barczynski et al.’s conclusions in their work on transient paralysis [16]. In 2017, in a meta-analysis based on 23 studies (9203 patients, 17,203 RLNs at risk), Yang et al. showed that the use of IONM during thyroid surgery significantly reduces the rates of total vocal fold paralysis and of transient vocal fold paralysis, although they failed to show statistical significance for reductions in the rate of permanent paralysis [17]. In another paper from 2017, Wonga et al. confirmed the superiority of IONM over visualization alone, but only in procedures with a higher risk of complications, namely reoperations and thyroid cancer. The percentages of all types of paralysis, as well as transient paralysis in patients with a higher risk of complications in the IONM group, were 4.5% and 3.9% vs. 2.5% and 2.4%, respectively, in the group with visualization alone, and the differences observed were at the level of statistical significance (*p* = 0.003; *p* = 0.016) [18]. The most favorable meta-analysis, by Bai and Chen from 2018, appears to demonstrate the superiority of IONM over VA in thyroid surgery [19]. The authors of this comprehensive meta-analysis, based on 34 publications from 1980–2017, showed a statistically significant decrease in all vocal fold paralysis (*p* = 0.0002), transient paralysis (*p* = 0.0003) and permanent paralysis (*p* = 0.0003) cases when comparing IONM to VA. They showed a statistically significant advantage of IONM over VA in thyroid cancer surgeries in terms of total paralysis (*p* = 0.02) and transient paralysis (*p* = 0.04); however, they did not show statistical significance for permanent paralysis (*p* = 0.73) [19].

The publications discussed above prove that demonstrating the absolute superiority of IONM over VA solely by directly comparing the rates of transient and permanent paralysis may be difficult for a number of reasons. If the primary endpoint of a study is only transient or permanent paralysis, it must be realized that the number of these events is very small, particularly at experienced hands of high-volume surgeons. Sanabria et al. showed that to achieve sufficient test power in randomized trials to confirm the superiority of IONM over VA in the prevention of RLN injury, it is necessary to analyze a minimum of 9000 RLNs at risk [25]. In contrast, Dralle et al. calculated that observation of a minimum of 39,907 nerves at risk of injury is needed to demonstrate the superiority of IONM over VA among patients undergoing surgery for thyroid cancer [28]. It is difficult to disagree with Davey et al. that this low incidence of complications contributes to potential underestimation of the significant impact of IONM in protecting the RLN from iatrogenic injury, and at the same time limits the ability to demonstrate its benefit to the population [12]. The low incidence of vocal fold paralysis after a thyroidectomy also limits the feasibility of prospective randomized single-center studies evaluating the value of IONM in the most objective manner. This results in a need for systematic reviews and meta-analyses to collect a large group of patients, although even these publications often prove insufficient to demonstrate the superiority of IONM over VA [12,13,14,15,23,25,26].

The question of whether we can currently demonstrate the superiority of IONM over visual identification therefore arises. It seems possible, provided we change our perspective on the value of IONM. After more than 10 years of experience working with neuromonitoring, the authors of this paper have noted that regardless of the indication for thyroid surgery, regardless of the type and extent of thyroid surgery and regardless of the current well-known risk factors for complications such as recurrent, giant, retrosternal or toxic goiters, surgeries performed with IONM were safe and had minimal complications in contrast to thyroid surgery with the above risk factors but without laryngeal nerve monitoring. We posited the rather bold hypothesis that the well-known risk factors for complications have no significant effect on the rate of vocal fold paralysis during thyroid surgery as long as it is performed with IONM, in contrast to their significant effect on the rate of vocal fold paralysis during surgery with only visual identification of the RLN. Proving this hypothesis from a practical point of view for the surgeon would undoubtedly prove the superiority of IONM over VA.

In order to verify the hypothesis, the authors of the current article retrospectively analyzed a group of 711 patients, of whom 257 were operated on with IONM (469 RLNs at risk) and the remaining 454 (796 RLNs at risk) with visual identification alone. In our first analysis, we showed that the total, transient and permanent number of vocal fold paralysis cases in the group of patients operated on with IONM was lower than in the group with VA, but not at the level of statistical significance (*p* = 0.2927; *p* = 0.6840; *p* = 0.229). These results came as no surprise and are in line with most meta-analyses conducted to date [12,13,14,15,23,26]. Although we did not find statistical significance, we showed a relative reduction of almost 30% in the total rate of vocal fold paralysis in operations performed with IONM vs. VA; for transient paralysis the reduction was 15%, and for permanent paralysis it was as high as 40%. In their extensive meta-analysis, Davey et al. showed more than a 25% reduction in all RLN injuries in surgeries using IONM; at the same time, those authors noted that one of the four instances of vocal fold paralysis could have been avoided if IONM had been used during thyroid surgery [12].

In subsequent analyses, we tried to prove the hypothesis that risk factors do not significantly affect the total rate of vocal fold paralysis in the group of patients operated on with IONM, in contrast to the patients with only VA, among whom risk factors significantly affect the increased rates of vocal fold paralysis.

In order to statistically evaluate the effect of neuromonitoring on the risk of postoperative complications in the two patient groups, two multigroup variable effect meta-analysis models were constructed using the odds ratio (OR) along with a 95% significance interval as the test metric. In both cases, the total risk of postoperative complications was determined by the values of 12 dichotomous variables known as risk factors.

In the first model of the meta-analysis, the chance of occurrence of one of the 12 risk factors analyzed was determined in the two groups of patients being compared (those in whom neuromonitoring was used during surgery and those in whom the only procedure used to identify the RLN was visualization). Our statistical analysis showed that in the group of patients in whom neuromonitoring was used, only one risk factor—the surgeon’s experience—proved statistically significant (OR = 3.27; *p* = 0.0478). In the group of patients where only visualization was used, the chances of 5 of the 12 factors analyzed were statistically significant: retrosternal goiter (OR = 2.23; *p* = 0.041), total volume (OR = 2.30; *p* = 0.0284), clinical diagnosis (OR = 2.5; *p* = 0.0669), gender (OR = 3.08; *p* = 0.0054) and risk stratification (OR = 3.30; *p* = 0.0041). In addition, the cumulative risk, taking into account the simultaneous influence of all 12 factors, was slightly higher in the group of patients in whom only visualization was used during the procedure: OR = 1.78. This value was also considerably more statistically significant (*p* < 0.0001) than that obtained in the group of patients in whom neuromonitoring was used: OR = 1.73; *p* = 0.004. Despite expectations, this meta-analysis showed no statistically significant differences between the cumulative OR values obtained in the two groups of patients.

The second meta-analysis model determined the chance of early postoperative complications in the group of patients who underwent a neuromonitoring procedure taking into account 1 of the 12 risk factors. This analysis showed that in each of the 12 subgroups, the chances of early postoperative complications among patients undergoing neuromonitoring were lower: the OR value for each of the 12 risk factors was less than 1. Although the individual OR values did not show statistical significance, the total chances of early postoperative complications taking into account the simultaneous influence of all 12 factors reached statistical significance: OR = 0.70; *p* = 0.016.

The statistical analyses performed clearly indicate that the use of the neuromonitoring procedure results in a noticeable reduction in the chances of various risk factors having an impact on the occurrence of postoperative complications. However, this fact, due to the large disproportion in the sizes of the groups of patients subjected only to visualization and those in whom neuromonitoring was used, is difficult to prove in a statistically unambiguous manner within the framework of clinical single-center studies. The observable correlations between the use of neuromonitoring, reduced chances of postoperative complications and the lower impact of potentially present risk factors on the final outcome of the procedure are also confirmed by the multivariate principal component analysis (PCA).

These results confirmed our hypothesis that commonly known risk factors for complications did not have a significant effect on the rate of vocal fold paralysis among patients operated on with IONM, while they still had a significant effect on increasing the prevalence of neural injuries in patients operated on with visual RLN identification only. Although the analyses based on risk factors were conducted only for the total number of instances of paralysis, this is undoubtedly strong indirect evidence of the superiority of IONM over VA in thyroid surgery. In clinical practice, this means that for male patients, procedures with a higher risk of complications (due to thyroid cancer or Grave’s disease), goiter volumes above 40 mL and retrosternal goiters, IONM monitoring of the RLN is warranted and should be routinely recommended. What is more, the use of neuromonitoring should be mandatory in selected groups of patients working professionally with voice, e.g., teachers [29].

This study has several limitations: (1) retrospective nature, (2) single-center and non-randomized design and (3) medium size of the cohort (but strong enough to allow for a multivariable analysis). On the other hand, the following strengths of the study should be acknowledged: (1) completeness of prospectively collected data and (2) inclusion into analysis of both low-volume and high-volume surgeons, reflecting the real-world landscape of thyroid surgery practice.

To summarize, the purpose of our study was to highlight the changing perception of the value of IONM in the prevention of RLN injury during thyroid surgery. Direct comparisons of the absolute number of cases of vocal fold paralysis among patients operated on with IONM vs. VA cannot currently be expected to attain statistical significance and may adversely affect assessments of the value of IONM in minimizing vocal fold paralysis for years to come. We hope that the hypothesis we have proven will change perceptions of the value of IONM, showing its superiority over surgery with only VA.

## 5. Conclusions

Risk factors for complications in thyroid surgery with IONM are not significant for any increase in the rate of vocal fold paralysis as long as surgery is performed with IONM, in contrast to thyroid surgery performed only with VA, thus proving the superiority of IONM over VA for safety.

## Figures and Tables

**Figure 1 biomedicines-11-00880-f001:**
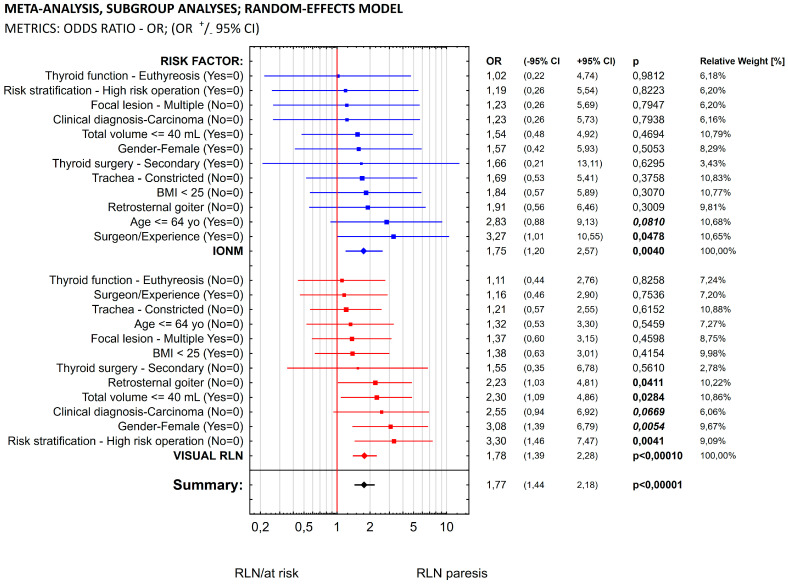
Meta-analysis, subgroup analyses (IONM vs. VA) and random effects model.

**Figure 2 biomedicines-11-00880-f002:**
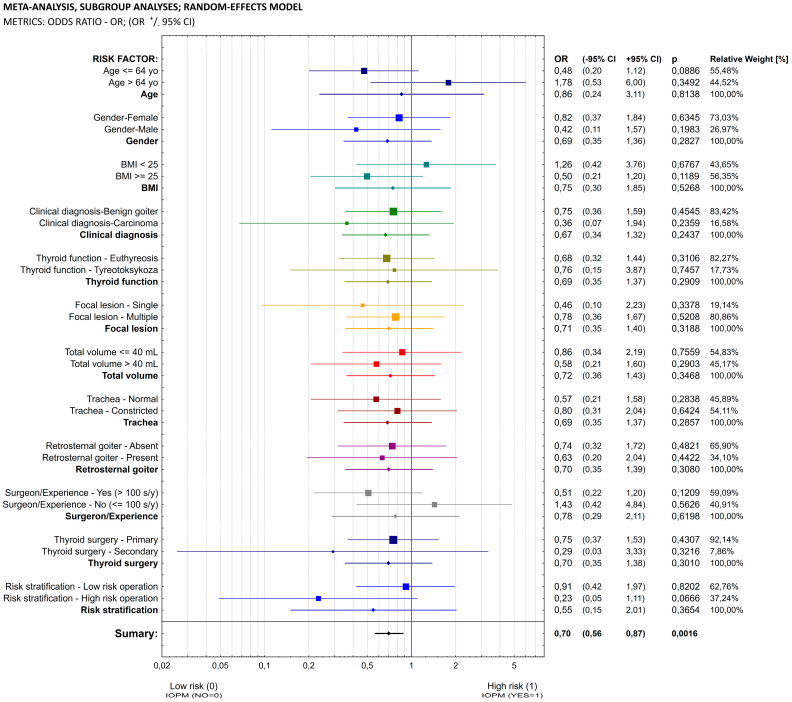
Meta-analysis, subgroup analyses (IONM vs. VA) and random effects model.

**Table 1 biomedicines-11-00880-t001:** Demographic data and clinical characteristics of patients operated with IONM vs.visual RLN identification alone.

Thyroid Operation	Intraoperating Nerve Monitoring	Visualization Alone
Number of patients, *n* (%)	257 (100%)	454 (100%)
Number of RLNs at risk of injury (RLN/at risk)	469	796
Age, mean ± standard deviation	53.4 ± 13.39	53.11 ± 14.79
Median	53.4	53.11
Minimum/maximum age (years)	17/86	17/82
Gender (female: male)	212:45	385:69
BMI, mean ± standard deviation (kg/m^2^)	26.79 ± 4.29	25.61 ± 3.13
Median	26.7	25.8
BMI minimum/maximum	16.6/45.78	12.9/35.9
Thyroid operation: primary, *n* (%)	220 (86%)	433 (95%)
Secondary, *n* (%)	37 (14%)	21 (5%)
Diagnosis, *n* (%)		
Benign goiter	221 (86%)	421 (7%)
Nodular goiter	182 (71%)	326 (72%)
Toxic nodular goiter	27 (11%)	76 (17%)
Graves’ disease	12 (5%)	19 (4%)
Thyroid’s carcinoma	36 (14%)	33 (7%)
Papillary	31 (12.5%)	29 (6.4%)
Follicular	4 (1.6%)	2 (0.4%)
Medullary	0	2 (0.4%)
Anaplastic	1 (0.4%)	0
Tracheal displacement/constriction, *n* (%)	115 (45%)	182 (40%)
Retrosternal goiter, *n* (%)	52 (20%)	96 (21%)
Thyroid volume (V), mean ± standard deviation	38.88 ± 31.3	38.29 ± 25.47
Median	29.7	32
V minimum/maximum (ml)	3.8/208.89	5/210
Thyroid surgery *n* (%)		
Total thyroidectomy	191 (74.32%)	270 (59.47%)
Lobectomy with the isthmus	45 (17.51%)	112 (24.67%)
Near-total thyroidectomy	13 (5.06%)	53 (11.67%)
Dunhill procedure	5 (1.95%)	11 (2.42%)
Subtotal bilateral thyroidectomy	3 (1.17%)	8 (1.76%)
Surgeon’s experience: low	56 (22%)	81 (18%)
High	201 (78%)	373 (82%)

**Table 2 biomedicines-11-00880-t002:** Risk factors for complications of vocal cord paralysis: dichotomous variables of meta-analyses. a Benign goiter (nodular goiter, toxic nodular goiter and Graves’ disease); b euthyroidism; c thyrotoxicosis; d thyroid volume (volume of one lobe = length × depth × width × π/6; e, f based on X-rays of the neck and chest; g low-risk operation (nodular goiter and toxic nodular goiter); h high-risk operation (Graves’ disease and carcinoma).

N	Variable	Subgroup
1	Age	Age ≤ 64 years
		Age > 64 years
2	Gender	Gender—female
		Gender—male
3	BMI	BMI < 25 (kg/m^2^)
		BMI ≥ 25 (kg/m^2^)
4	Clinical diagnosis ^a^	Benign goiter
		Carcinoma
5	Thyroid function	Euthyroidism ^b^
		Thyrotoxicosis ^c^
6	Focal lesion	Single
		Multiple
7	Total volume ^d^	≤40 mL
		>40 mL
8	Trachea ^e^	Normal
		Constricted
9	Retrosternal goiter ^f^	Absent
		Present
10	Surgeon experience	High volume (>100 s/y)
		Low volume (≤100 s/y)
11	Thyroid surgery	Primary
		Secondary
12	Risk stratification	Low-risk operation ^g^
		High-risk operation ^h^
13 *	Extent of thyroid surgery	Total thyroid lobectomy
		Partial thyroid lobectomy

* Risk factor not included in meta-analyses (paresis with IONM-0).

**Table 3 biomedicines-11-00880-t003:** The total, transient and permanent number of occurrences of vocal fold paralysis in the group of patients operated on with IONM vs. VA.

Paralysis of the Vocal Cords	Overall RLN at Risk *n* (%) 1265 (100%)	IONM RLN at Risk *n* (%) 469 (100%)	VA RLN at Risk *n* (%) 796 (100%)	Pearson’s chi^2^ Test	*p*	OR	“–95% CI”	“+95% CI”
Overall VCP	41 (3.24%)	12 (2.56%)	29 (3.64%)	1.1069	0.2927	0.69	0.35	1.37
Transient VCP	27 (2.13%)	9 (1.92%)	18 (2.26%)	0.1655	0.6840	0.84	0.37	1.89
Permanent VCP	14 (1.11%)	3 (0.64%)	11 (1.38%)	1.4855	0.2229	0.45	0.12	1.65

**Table 4 biomedicines-11-00880-t004:** Risk factors and vocal cord paralysis in thyroid operation with IONM vs. visual RLN identification alone—Pearson’s chi^2^ test.

Risk Factor for Vocal Cord Paralysis		IONM		VA		IONM vs. VA				
		RLN at Risk n (%)	RLN Paresis *n* (%)	RLN at Risk *n* (%)	RLN Paresis *n* (%)	Pearson’s chi^2^ Test	*p*	OR	“−95% CI”	“+95% CI”
Thyroid surgery	Primary	408	11 (2.7%)	759	27 (3.56%)	0.6247	0.4307	0.75	0.37	1.53
	Secondary	61	1 (1.64%)	37	2 (5.4%)	1.1007	0.3216	0.29	0.03	3.33
Age	<65	372	7 (1.88%)	593	23 (3.88%)	3.0261	0.0886	0.48	0.20	1.12
	≥65	97	5 (5.15%)	203	6 (2.96%)	0.8985	0.3492	1.78	0.53	6.00
Gender	Female	386	9 (2.33%)	674	19 (2.82%)	0.2267	0.6345	0.82	0.37	1.84
	Male	83	3 (3.61%)	122	10 (8.2%)	1.7462	0.1983	0.42	0.11	1.57
BMI	<25 (kg/m^2^)	133	5 (3.76%)	333	10 (3.0%)	0.1745	0.6767	1.26	0.42	3.76
	≥25 (kg/m^2^)	336	7 (2.08%)	463	19 (4.1%)	2.5244	0.1189	0.50	0.21	1.20
Clinical diagnosis	Benign goiter	403	10 (2.48%)	733	24 (3.27%)	0.5629	0.4545	0.75	0.36	1.59
	Thyroid carcinoma	66	2 (3.03%)	63	5 (7.94%)	1.5118	0.2359	0.36	0.07	1.94
Clinical diagnosis/risk stratification	Low-risk operation	379	10 (2.64%)	695	20 (2.89%)	0.0516	0.8202	0.91	0.42	1.97
	High-risk operation	90	2 (2.22%)	101	9 (8.91%)	3.9229	0.0666	0.23	0.05	1.11
Thyroid function	Euthyroidism	392	10 (2.55%)	618	23 (3.72%)	1.0400	0.3106	0.68	0.32	1.44
	Thyrotoxicosis	77	2 (2.6%)	178	6 (3.37%)	0.1057	0.7457	0.76	0.15	3.87
Focal lesion	Single	94	2 (2.17%)	175	8 (4.57%)	0.9614	0.3378	0.46	0.10	2.23
	Multiple	377	10 (2.65%)	621	21 (3.38%)	0.4143	0.5208	0.78	0.36	1.67
Total volume	≤40 mL	319	7 (2.19%)	513	13 (2.53%)	0.0967	0.7559	0.86	0.34	2.19
	≥40 mL	150	5 (3.33%)	283	16 (5.65%)	1.1438	0.2903	0.58	0.21	1.60
Extent of thyroid surgery	Total thyroid lobectomy	458	12 (2.62%)	769	28 (3.64%)	0.9488	0.3300	0.71	0.35	1.41
	Partial thyroid lobectomy	11	0	27	1 (3.7%)	0.4184	0.5177	-	-	-
Trachea	Normal	255	5 (1.96%)	475	16 (3.37%)	1.1767	0.2838	0.57	0.21	1.58
	Displaced/constricted	214	7 (3.27%)	321	13 (4.05%)	0.2164	0.6424	0.80	0.31	2.04
Retrosternal goiter	Absent	370	8 (2.16%)	620	18 (2.9%)	0.4976	0.4821	0.74	0.32	1.72
	Present	99	4 (4.04%)	176	11 (6.25%)	0.5998	0.4422	0.63	0.20	2.04
Surgeon experience	≤100 surgeries/year	87	5 (5.75%)	147	6 (4.08%)	0.3384	0.5626	1.43	0.42	4.84
	>100 surgeries/year	382	7 (1.83%)	649	23 (3.54%)	2.4931	0.1209	0.51	0.22	1.20

**Table 5 biomedicines-11-00880-t005:** Influence of risk factors for vocal cord paralysis in visual RLN identification alone—Pearson’s chi^2^ test.

Risk Factor for Vocal Cord Paralysis		IONM							VA						
		RLN at Risk *n* (100%)	RLN Paresis *n* (%)	Pearson’s chi^2^ Test	*p*	OR	“−95% CI”	“+95% CI”	RLN at Risk *n* (%)	RLN Paresis *n* (%)	Pearson’s chi^2^ Test	*p*	OR	“−95% CI”	“+95% CI”
Thyroid surgery	Primary	408	11 (2.70%)	0.2376	0.6295	1.66	0.21	13.11	759	27 (3.56%)	0.3432	0.5610	1.55	0.35	6.78
	Secondary	61	1 (1.64%)						37	2 (5.41%)					
Age	<65 years	372	7 (1.88%)	3.3056	0.0810	2.83	0.88	9.13	593	23 (3.88%)	0.3669	0.5459	1.32	0.53	3.30
	≥65 years	97	5 (5.15%)						203	6 (2.96%)					
Gender	Female	386	9 (2.33%)	0.4509	0.5053	1.57	0.42	5.93	674	19 (2.82%)	8.5101	0.0054	3.08	1.39	6.79
	Male	83	3 (3.61%)						122	10 (8.20%)					
BMI	<25 (kg/m^2^)	133	5 (3.76%)	1.0736	0.3070	1.84	0.57	5.89	333	10 (3.00%)	0.6684	0.4154	1.38	0.63	3.01
	≥25 (kg/m^2^)	336	7 (2.08%)						463	19 (4.10%)					
Clinical diagnosis	Benign goiter	403	10 (2.48%)	0.0685	0.7938	1.23	0.26	5.73	733	24 (3.27%)	3.5922	0.0669	2.55	0.94	6.92
	Thyroid carcinoma	66	2 (3.03%)						63	5 (7.94%)					
Clinical diagnosis/risk stratification	Low-risk operation	379	10 (2.64%)	0.0505	0.8223	1.19	0.26	5.54	695	20 (2.88%)	9.1436	0.0041	3.30	1.46	7.47
	High-risk operation	90	2 (2.22%)						101	9 (8.91%)					
Thyroid function	Euthyreosis	392	10 (2.55%)	0.0005	0.9812	1.02	0.22	4.74	618	23 (3.72%)	0.0484	0.8258	1.11	0.44	2.76
	Thyrotoxicosis	77	2 (2.60%)						178	6 (3.37%)					
Focal lesion	Single	92	2 (2.17%)	0.0679	0.7947	1.23	0.26	5.69	175	8 (4.57%)	0.5505	0.4598	1.37	0.60	3.15
	Multiple	377	10 (2.65%)						621	21 (3.38%)					
Total volume	≤40 mL	319	7 (2.19%)	0.5308	0.4694	1.54	0.48	4.92	513	13 (2.53%)	5.0561	0.0284	2.30	1.09	4.86
	≥40 mL	150	5 (3.33%)						283	16 (5.65%)					
Extent of thyroid surgery	Total thyroid lobectomy	458	12 (2.62%)	0.2957	0.5865	-	-	-	769	28 (3.64%)	0.0002	0.9863	0.98	0.12	7.50
	Partial thyroid lobectomy	11	0						27	1 (3.70%)					
Trachea	Normal	255	5 (1.96%)	0.8011	0.3758	1.69	0.53	5.41	475	16 (3.37%)	0.2533	0.6152	1.21	0.57	2.55
	Displaced/constricted	214	7 (3.27%)						321	13 (4.05%)					
Retrosternal goiter	Absent	370	8 (2.16%)	1.1051	0.3009	1.91	0.56	6.46	620	18 (2.90%)	4.3739	0.0411	2.23	1.03	4.81
	Present	99	4 (4.04%)						176	11 (6.25%)					
Surgeon/ Experience	≤100 surgeries/year	87	5 (5.75%)	4.3555	0.0478	3.27	1.01	10.55	147	6 (4.08%)	0.0987	0.7536	1.16	0.46	2.90
	>100 surgeries/year	382	7 (1.83%)						649	23 (3.54%)					

## Data Availability

The datasets used and/or analyzed during the current study are available from the corresponding author upon reasonable request.

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
