# Peer review of "Proving the Superiority of Intraoperative Recurrent Laryngeal Nerve Monitoring over Visualization Alone during Thyroidectomy"

_biomedicines, 2023, doi:10.3390/biomedicines11030880_

Round 1

Reviewer 1 Report

Good work but without new informations. The importance of the IONM is especially for particular professional category (e.g. teachers). ABout this you should mention this manuscript:

Galletti B, Sireci F, Mollica R, Iacona E, Freni F, Martines F, Scherdel EP, Bruno R, Longo P, Galletti. Vocal Tract Discomfort Scale (VTDS) and Voice Symptom Scale (VoiSS) in the Early Identification of Italian Teachers with Voice Disorders. Int Arch Otorhinolaryngol. 2020;24(3):e323-e329. doi:10.1055/s-0039-1700586

Author Response

Response to Reviewer 1 Comments.

Dear Reviewer, 

Thank you for taking the time to read my manuscript and for any comments. I agree with you that the use of IONM is especially usefull for particular professional category (e.g. teacher). About this I mentioned in Discussion according to Galletti et all. publication ( I have added References 29).

All changes are in red in manuscript. 

Reviewer 2 Report

The authors described a important problem in thyroid surgery. 

the cohorte is described with low relevancy

no risk factor definition based on evidence 

no defintion of exprienced and non experienced surgeon to the relevan literature

no defintion af age defintion for study results

no defintion of choice of Va I IONM and c IONM

no definition of risk of surgery total hemi partiel near thzroidectomy to nerve at risk

no definition of risk syrgery groups

no explanation of nerve at risk no neck lymph node dissection by cancer with vagak nerves at risk

no defintion of the period for decision for temporaraz vocal cord palsy 

no defintion of nerve palsy diagnostic set 

no period of observation of patients after surgery 

Author Response

Thank you for all comments and suggestions.

Below are my explanations and corrections.

Point 1. the cohorte is described with low relevancy

Response 1.

I hope that the following additions will make the cohort better described.

Point 2. no risk factor definition based on evidence 

Response 2.

Risk factors in thyroid surgery are widely known, but there is no single definition. In addition, the authors of the publication tried to list as many factors as possible that could have a potential impact on complications. All risk factors were redefined.

I added in link: 149-172

Risk factors definitions:

  • Age definition: Age <= 64 yo - patients up to 64 years of age/Age > 64 yo - patients aged 65 and over
  • Gender: Female, Male
  • BMI: BMI < 25 [kg / m²], BMI >= 25 [kg / m²]

4) Clinical diagnosis: Benign goiter (nodular goiter , toxic nodular goiter, Graves’ disease)/ Carcinoma (papillary, follicular, medullary, anaplastic ). Patients with thyroid cancer underwent thyroid operation with central lymph node dissection.

5) Thyroid function: Euthyroidism (patients with nodular goiter without thyroid dysfunction before surgery)/ Thyrotoxicosis (patient treated before thyroid operation due to thyrotoxicosis; patients with toxic nodular goiter, Graves disease)

6) Focal lesion: Single (single thyroid nodule)/ Multiple (multiple thyroid nodule) according to ultrasound examination

7) Total volume (the volume of the thyroid lobe determined on the basis of the formula for the volume of the volume of one lobe = length × depth × width × π/6: <= 40 mL ( thyroid volume up to 40 ml), > 40 mL (thyroid volume from 40 ml and above)

8) Trachea (based on x-rays of the neck and ches): Normal/ Constricted

9) Retrosternal goiter (based on x-rays of the neck and ches): Absent/Present

10) Surgeon/Experience: High Volume (> 100 s/y)- Surgeon performing up to 100 operations per year/ Low Volume (<= 100 s/y)- Surgeon performing over 100 operations a year

11) Thyroid surgery: Primary (First thyroid operation on the thyroid gland/Secondary (recurrent goiter, radicalization due to cancer)

12) Risk stratification: Low risk operation(patients operated due to nodular goiter and toxic nodular goiter)/High risk operation (patient operated due to Graves’ disease and thyroid cancer)  

13) Extent of thyroid surgery: Total thyroid lobectomy ( the entire thyroid lobe is removed)/Partial thyroid lobectomy ( a partial of thyroid lobe is removed)

Point 3. no defintion of exprienced and non experienced surgeon to the relevan literature

Response 3.

There is no single universally accepted definition of an experienced/inexperienced surgeon. This is the subject of many works and definitions vary. Often in endocrine surgery, the conclusions of the papers are that a minimum of 100 thyroid surgeries/year is needed to gain experience.

I added link 165-166.

10) Surgeon/Experience: High Volume (> 100 s/y)- Surgeon performing up to 100 operations per year/ Low Volume (<= 100 s/y)- Surgeon performing over 100 operations a year

Point 4 no defintion af age defintion for study results

Response 4.

I added in link 149:

  • Age definition: Age <= 64 yo - patients up to 64 years of age/Age > 64 yo - patients aged 65 and over

Point 5. no defintion of choice of Va I IONM and c IONM

Response 5.

We did not use c-IONM, hence the lack of information about this type of surgery in study group.

I added link 115-117:

For the first 4 months of each year, patients were operated on with IONM, in the following months operations were performed without IONM, only with VA.

Point 6. no definition of risk of surgery total hemi partiel near thzroidectomy to nerve at risk

Response 6.

I added in link 171-172:

13) Extent of thyroid surgery: Total thyroid lobectomy ( the entire thyroid lobe is removed)/Partial thyroid lobectomy ( a partial of thyroid lobe is removed)

Point 7. no definition of risk syrgery groups

Response 7.

I added in Link 169-170:

12) Risk stratification: Low risk operation(patients operated due to nodular goiter and toxic nodular goiter)/High risk operation (patient operated due to Graves’ disease and thyroid cancer) 

Point 8.  explanation of nerve at risk no neck lymph node dissection by cancer with vagak nerves at risk

Response 8.

I added in link 152-154:

4) Clinical diagnosis: Benign goiter (nodular goiter , toxic nodular goiter, Graves’ disease)/ Carcinoma (papillary, follicular, medullary, anaplastic ). Patients with thyroid cancer underwent thyroid operation with central lymph node dissection.

Point 9. no defintion of the period for decision for temporaraz vocal cord palsy 

Response 9.

The definition of transient or permanent paralysis is in line 181-183:

Transient vocal fold paralysis (T-VCP) was defined as paralysis resolving up to 12 months after thyroid surgery; permanent paralysis (P-VCP) was defined as paralysis persisting 12 months and more after surgery.

Point 10. no defintion of nerve palsy diagnostic set 

Response 10.

I added in link: 178-181:

To assess the mobility of the vocal folds, both preoperatively and postoperatively (2 days to 2 weeks after surgery, and one year later) each patient underwent an ENT examination (video laryngoscopy or indirect laryngoscopy) to assess vocal fold mobility.

Point 11. no period of observation of patients after surgery.

Response 11.

I added in line 175-176 (in red):

 All patients underwent thyroid operation were followed for 12 months after surgery to assess complications.

Kind regards, 

Beata Wojtczak

Round 2

Reviewer 2 Report

The authors described the inclusions criterias and risk factors as a specific rules for this study only. The description of age 64 years is incoherent to literature. The methods of palsy of recurrent nerve without the laryngoscopy and voice analyse and or videkxmugraphy is not objective. The time of temporary palsy were not described widely. The incidence of palsy is described to the exposed nerves without the uni or bilateral kncidence , incidency by revision surgery and incidence of palsy by neck dissection by cancer patients. The analyse is full by sratistical data without the statistical analyse of incidence in literature. The authors used the special therms of experienced surgeon and unexperienced surgeon without the respect to the exist literature. The eyplanation of the using of personal data evidence without the compareto the exisr literature is going to the uncomletly analyse and the concluisions are unspecific. The main ideais that the authors done a lot of thyroid procedures using the neuromonitoring. In this form is not possible recommand the printing of that. 

Author Response

Response to Reviewer 2 Comments- Round 2

Ad 1. The authors described the inclusions criterias and risk factors as a specific rules for this study only.

The inclusion criteria:  From Manuscript: …..The medical records of 711 patients (1265 recurrent laryngeal nerves at risk of injury) who were surgically treated for various disorders of the thyroid gland at the Department of General, Gastroenterological and Endocrine Surgery of the Medical University of Wroclaw in 2012-2015 were analyzed retrospectively ……In 257 patients (469 RLNs at risk), thyroid surgery was performed with intraoperative neuromonitoring (IONM), and in 454 patients (796 RLNs at risk), only visual identification of the RLN was used during thyroid surgery…..

This fragment shows that all patients operated on in the given period (both with and without neuromonitoring) were included in the study.There was no mention in the text of the factors of inclusion or exclusion from the study.

risk factors as a specific rules for this study only-  As I mentioned in Response to reviewer 2 (Round 1)  risk factors in thyroid surgery are widely known, but there is no single definition. All thyroid’s or endocrine’s surgeons know these risk factors. Moreover all publications about the risk factors in thyroid surgery are the same as in my publication.  In addition, the authors of the publication tried to list as many factors as possible that could have a potential impact on complications. All risk factors were detailed redefined in Response for a Reviever 2 in Round 1.

Ad 2. The description of age 64 years is incoherent to literature.

More than 2000 medical publications in Pub Med shows that age 65 is connected with old age. Of course, life expectancy is getting longer and there are various definitions of old age, although the limit of 65 years and above is still mostly used for describing old age and this is how it was adopted in my publication. So, I divided the study group to 64 old year and 65 old year and over.

Ad 3. The methods of palsy of recurrent nerve without the laryngoscopy and voice analyse and or videkxmugraphy is not objective.

I agree that in order to objectively assess paralysis, it is necessary to perform laryngoscopy – and it was done in my study at the beginning – it was in the first version of my manuscript and I have written about it in Round 1: In my last version of manuscript there is Link 175-178:

To assess the mobility of the vocal folds, both preoperatively and postoperatively (2 days to 2 weeks after surgery, and one year later) each patient underwent an ENT examination (video laryngoscopy or indirect laryngoscopy) to assess vocal fold mobility.  

voice analyse--- the analysis of voice evaluation is only a subjective examination and as you know in 30% of patients after thyroidectomy with paresis of the vocal folds  (due to RLN palsy) may be without any symptoms – it means they have a good voice. Yes, of course it is important in patients with external branch of the superior laryngeal (EBSLN) nerve injury - but EBSLN palsy was not the topic of this research.

videkxmugraphy -  I’m sorry, I don’t understand. Maybe there is a linguistic error here . If you thought about electromyography of the larynx – it is only relevant for the assessment of EBSLN injury and as I said it was not the subject of this study. 

Ad 4: The time of temporary palsy were not described widely

It is not true. From the very beginning (in the first version and in Round 1) , transient and permanent paralysis was precisely defined

In actual manuscript Link 178-180:

Transient vocal fold paralysis (T-VCP) was defined as paralysis resolving up to 12 months after thyroid surgery; permanent paralysis (P-VCP) was defined as paralysis persisting 12 months and more after surgery.

Ad 5. The incidence of palsy is described to the exposed nerves without the uni or bilateral kncidence , As you know, there are different methods of presentation of vocal fold paralysis in the literature. We evaluate RLN palsy as a rate of RLN at risk of injury, so we did not evaluate uni or bilateral paresis for an obvious reason- it is not possible to asses the risk factor in this way. Moreover- as you know bilateral paresis prevalence is less than 1% - it doesn't have any statistical power, so all the paresis in such kind of study are presented as a rate of RLN at risk, as in similar publications.

 incidency by revision surgery and incidence of palsy by neck dissection by cancer patients- Secendary thyroid operation was -- all secondary operations have been included in one group, separating them into individual subgroups means that their numbers are small and it is impossible to calculate the statistics.

 Ad 6 The analyse is full by sratistical data without the statistical analyse of incidence in literature.

Not true. A broad reference to the literature is in the discussion.

Ad 7 The authors used the special therms of experienced surgeon and unexperienced surgeon without the respect to the exist literature.

As I said in (Response 3 -Round 1).

There is no single universally accepted definition of an experienced/inexperienced surgeon. This is the subject of many works and definitions vary. Often in endocrine surgery, the conclusions of the papers are that a minimum of 100 thyroid surgeries/year is needed to gain experience.

I added link 162-163.

10) Surgeon/Experience: High Volume (> 100 s/y)- Surgeon performing up to 100 operations per year/ Low Volume (<= 100 s/y)- Surgeon performing over 100 operations a year

Ad 8. The eyplanation of the using of personal data evidence without the compareto the exisr literature is going to the uncomletly analyse and the concluisions are unspecific.

Sorry, I do not understand.

Ad 9. The main ideais that the authors done a lot of thyroid procedures using the neuromonitoring.

What is written above is not true. This is not the idea of this work. The idea of this study is quite different. The idea of this study was to attempt to prove the superiority of IONM over VA of the RLN during thyroid surgery in the prevention of vocal fold paralysis; taking into account risk factors for complications.

Best Regards,

Beata Wojtczak MD, PHD
